# Impacts of Cross-Sectoral Climate Policy on Forest Carbon Sinks and Their Spatial Spillover: Evidence from Chinese Provincial Panel Data

**DOI:** 10.3390/ijerph192114334

**Published:** 2022-11-02

**Authors:** Hongge Zhu, Yingli Cai, Hong Lin, Yuchen Tian

**Affiliations:** 1College of Economics and Management, Northeast Forestry University, Harbin 150040, China; 2College of Marxism, Minjiang University, Fuzhou 350108, China

**Keywords:** climate change, cross-sectoral coordination, climate policy, forest carbon sinks

## Abstract

This paper examines the impact of cross-sectoral climate policy on forest carbon sinks. Due to the complexity of the climate change issue and the professional division of labor among government departments, cross-sectoral cooperation in formulating climate policy is a desirable strategy. Forest carbon sinks play an important role in addressing climate change, but there are few studies focusing on forest carbon sinks and cross-sectoral climate policies. Thus, based on the panel data of 30 provinces and cities in China from 2007 to 2020, this paper establishes a benchmark regression model and a spatial panel model to analyze the impact of cross-sectoral climate policies on forest carbon sinks. We find that cross-sectoral climate policies positively impact forest carbon sinks. Under the influence of the “demonstration effect”, we find that cross-sectoral climate policies have a positive impact not only on the forest carbon sinks in the region but also on those in the neighboring region. Further analysis shows that for provinces with less developed forestry industry and small forest areas, the positive effect of cross-sectoral climate policies on forest carbon sinks is more obvious. Overall, this paper can serve as an important reference for local governments to formulate climate policies and increase the capacity of forest carbon sinks.

## 1. Introduction

Climate change has become an important challenge for human development in the 21st century and one of the most important issues in global development [1,2,3]. Climate change is a problem that transcends national borders [4], and there is a consensus that addressing climate change is a responsibility shared by all countries [5]. In 2007, China surpassed the United States as the world’s largest emitter of greenhouse gases (GHG), currently accounting for about 30% of global emissions [6,7]. Faced with the pressures of climate change, action is required by the Chinese government for mitigation. In June 2007, the Chinese government established the “National Leading Group on Climate Change” and later issued the “National Climate Change Program” (NCCP). Against the backdrop of immense volumes of GHG emissions and national policies to address climate change, China’s provincial governments have developed provincial climate policies in response to the central government’s requirements for climate change mitigation and adaptation. The implementation of climate policies not only enables the main benefits of climate warming mitigation but also generates auxiliary benefits in regional economic development, ecological environmental protection, social stability, and other aspects [8]. Apart from these benefits, natural ecosystems are also significantly impacted by regulating actions, such as regarding land use type transformation and forest management [9,10]. It is worth noting that the forest, because of its high carbon storage capacity and productivity, has attracted attention in the context of climate change mitigation [11]. Thus, exploring the issue of forest carbon sinks from a climate policy perspective is not only a requirement for clarifying the effects of climate policies but also has important practical implications.

Forest carbon sinks, which currently represent the most cost-effective way to reduce emissions, play an important role in mitigating climate change. Specifically, the activities of forest carbon sinks in vegetation and soil can be affected through forest management and vegetation restoration, using the process of plant photosynthesis to absorb carbon dioxide from the atmosphere [5]. Since emission reduction methods involving forest carbon sinks have the characteristics of being cheaper and more efficient than others, such as industrial emission reduction [12], the use of forest carbon sinks is becoming an important strategy to reduce carbon emissions. Moreover, forest product revenues can be used to hedge some of the costs in the process of forest carbon sink construction [13]. In addition, forest carbon sinks are also crucial for neutralizing the residual carbon emissions from other sectors, such as the industrial and agricultural sectors [14]. If the climate policy planned by government departments does not include relevant measures, such as improving the capability of forest carbon sinks and achieving the goal of tackling climate change, this will result in increased policy costs [15]. 

How to efficiently improve and increase forest carbon sinks is a hot issue in the current research on combating climate change. The influencing factors of forest carbon sinks involve fields both in natural and social sciences, which are closely related. From the perspective of natural factors, changes in climate and atmospheric factors (CO_2_, nitrogen deposition, ozone, etc.) will affect forest carbon sinks [16]. Regarding forests, a reduction in deforestation and an increase in forest area both contribute to a significant increase in forest carbon sinks [17]. From the point of socioeconomics, the factors of economic development, wood product consumption, population, income level, and land use structures significantly change the forest carbon sinks [3,13,18,19]. For government departments, improving the capacity of forest carbon sinks requires not only the support from forestry departments but also the coordination of office departments, financial support from finance departments, and technical support from science and technology departments. As a result, frequent interactions among many government departments will result in a unified policy jointly promulgated by numerous departments [20]. A policy that has a high degree of synergy is more likely to achieve the desired goal of increasing forest carbon sinks.

Previous studies on climate policy have focused on the evolutionary trends, evaluation of effects, and synergy with other policies [9,10,21,22,23,24,25,26]. In terms of policy content, climate policy is a comprehensive policy that includes climate change mitigation and adaptation and is integrated with local development [27]. Considering that climate policy is an important initiative to deal with climate change, the design of climate policy and the economic and environmental effects of climate policy is increasingly becoming objects of attention for government departments and research scholars in various countries [28]. A single sector cannot effectively address the complex climate change issue, and cross-sectoral climate policies are imperative [29,30,31]. This is because of the specialization in the division of labor within government departments and the complexity and systematicity of climate change [29]. Thus, the cross-sectoral climate policy mentioned in this paper refers to the process of coordinating between governments at the same level and between different functional departments of the same government regarding the formulation and implementation of climate policies to achieve the climate policy goals of more effectively addressing climate change. In other words, the cross-sectoral climate policy mentioned is a management decision of horizontal government integration [32]. Due to the game of administrative power, the coordination of interests among many subjects, and the requirements of responding to climate change, the involvement of all departments will strengthen the synergy in climate policy. So, can the current cross-sectoral climate policies of Chinese provincial governments contribute to the development of forest carbon sinks? Addressing this question not only helps in understanding the current status of cross-sectoral cooperation and governance in the field of climate change among Chinese provincial governments but will also result in the provision of empirical references for local governments to plan efficient climate policies and promote the effective increase of forest carbon sinks.

In summary, there is a wealth of research on forest carbon sinks and extensive research on climate policy. A review of the literature reveals that forest carbon sinks occupy an important position in the actions of countries around the world to address climate change [11], but few studies have analyzed forest carbon sinks in connection with government policies to address climate change. Moreover, while climate policy is primarily the task of national governments and international agreements and processes, the responsibility for implementing climate policy tends to fall on local governments [33]. Moreover, in the analysis of the influencing factors of forest carbon sinks, time series models, cross-section regression models, and panel regression models are currently being adopted, and less consideration has been given to the spatial dependence of forest carbon sinks. In the development of forest carbon sinks, however, achieving complete regional segmentation is difficult due to the natural properties of forest resources, and obvious regional correlation effects will manifest [13,34]. Hence, local governments must adhere to the principle of combining local management and regional collaboration in the pursuit of increasing forest carbon sinks, and the impact on forest carbon sinks of cross-sectoral coordination on climate policies among government departments at the same level deserves attention and testing. Thus, this paper not only uses a general panel model for regression but also considers the spatial correlation of forest carbon sinks between regions and the spatial spillover effect of cross-sectoral climate policies on forest carbon sinks, and spatial panel regression is used to analyze the relationship between cross-sectoral climate policies and forest carbon sinks. To ensure the availability and comprehensiveness of data, this paper takes 30 provinces in China from 2007 to 2020 as research samples. By studying the impact of cross-sectoral climate policies on forest carbon sinks, we can both enrich the selection of socioeconomic factors of forest carbon sinks and provide more accurate empirical references for further improving the capacity of forest carbon sinks.

The innovations and contributions of this paper are as follows: (1) The research perspective is novel. Few studies have explored the impact of cross-sectoral climate policies on forest carbon sinks from the perspective of climate policy despite there being many studies on the influencing factors of forest carbon sinks; (2) Empirical research findings in the frame of research on the influencing factors of forest carbon sinks and the connection with cross-sectoral climate policy and spatial factors. Based on Chinese provincial panel data, the spatial panel model is established based on empirical research on the impact of climate policy on forest carbon sinks across departments at the provincial level. Moreover, natural factors such as temperature and precipitation as well as socioeconomic factors such as economic development, urbanization, and energy structure are included in the control variables; (3) Based on the results of empirical analysis and the actual situation of regional forest carbon sinks, we propose targeted suggestions to improve the capacity of forest carbon sequestration. This has a certain inspirational effect on local government departments when planning climate policies and provides them with an important reference for promoting the development of forest carbon sinks.

## 2. Literature Review and Research Hypothesis

Due to the public goods nature of climate issues, climate protection practices are prone to “free-riding” [8,21]. At the same time, due to the decentralization of decision-making power and the difference in the interests of multiple subjects, the practice of climate policy is also susceptible to the influence of politics and stakeholders [35]. Moreover, the formulation of climate policy may represent a risk in terms of promoting competition for administrative resources and the expansion of power among particular government departments [36]. However, the complexity and systematicity of climate change issues determine the need for government departments to carry out cross-sectoral cooperation [2,37]. In this context, cross-sectoral climate policies carry out planning and comprehensive regulation for all government departments to secure more government support [20]. This can improve the quality of climate policies in the stages of policy target formulation, the determination of detailed policy rules, and policy implementation [29]. As a result, cross-sectoral climate policies may be more helpful in addressing climate change issues [38]. In China, climate policy mainly involves the departments of ecology and environment, energy, finance, and the Development and Reform Commission (DRC), while the essence of cross-sectoral climate policy is a synergistic process in the above departments.

### 2.1. Impact of Cross-Sectoral Climate Policies on Forest Carbon Sinks

Examining the influence of forest carbon sinks requires not only considering the driving role of natural factors but also the impact of policy initiatives, such as climate policy, on forest carbon sinks [39]. At the international level, Ge and Lin used a synthetic control method for analysis and found that the Kyoto Protocol had a significant promoting effect on the development of forest carbon sinks in signatory countries [40]. At the national and local government levels, climate policy has gradually shifted from single-sectoral to cross-sectoral policies [41]. As Hunt and Watkiss argued, even though climate change impacts may be primarily associated with a specific government department, they also have an intrinsic impact on many other interconnected government departments [42]. At the same time, cross-sectoral climate policy will be the mainstream of future climate policy because it creates synergies, or potential synergies, and facilitates multiple government departments in achieving their climate change mitigation goals [43]. Thus, it is necessary to explore the impact of cross-sectoral climate policy on forest carbon sinks from the perspective of climate policy. Cross-sectoral climate policy allows for the existence of many policy measures, and initiatives in multiple sectors reflect the demand for a more coordinated climate strategy [44]. Through empirical analysis, Xu et al. found that cross-sectoral climate policy is more conducive to achieving the goal of increasing forest carbon sinks by implementing multiple strategies rather than only a single strategy [45]. Moreover, cross-sectoral climate policies not only reap the benefits of reduced carbon emissions but also promote carbon emissions and carbon sequestration to offset each other [46]. It can be seen that a cross-sectoral climate policy can not only promote the increase of forest carbon sinks but also reduce the cost of climate policies. Accordingly, Hypothesis 1 is proposed in this paper.

**Hypothesis 1:** 
*Cross-sectoral climate policies can help increase forest carbon sinks.*


### 2.2. Spatial Spillover Effects of Cross-Sectoral Climate Policies on Forest Carbon Sinks

The natural attributes of forest resources make it difficult to realize the complete segmentation of forest carbon sinks in a region, leading to regional association effects. Thus, the spatial effect should be considered when analyzing the influencing factors of forest carbon sinks. According to Tobler’s first law of geography, objects with similar spatial distribution are more closely related to each other [47]. The empirical test showed that forest carbon sinks had a significant spatial correlation, whether at the national or provincial level [13,34]. In China, the regions with the largest forest carbon sinks are mainly located in southwest and northeast Asia [48], and there are significant clustering characteristics of forest carbon sinks between regions [34]. It can be seen that when the forest carbon sinks in this region change, the forest carbon sinks in neighboring regions will also change. From the perspective of climate policy, a cross-sectoral climate policy can promote the increase in forest carbon sinks by providing incentives or establishing relevant laws and regulations [49]. Cross-sectoral climate policy can, in addition, avoid the overlap of activities aimed at increasing forest carbon sinks as much as possible, thus achieving the integration and coordination of interests among subjects of climate policy [50]. Cross-sectoral climate policies that coordinate the interests of various actors may have a “demonstration effect”. Regions implementing cross-sectoral climate policies can achieve synergy and contribute significantly to forest carbon sinks while achieving climate change mitigation. Cross-sectoral synergies between government departments in implementing climate policies may be imitated and strengthened in neighboring regions, thereby promoting the development of forest carbon sinks in the region. Thus, this paper puts forward Hypothesis 2.

**Hypothesis 2:** 
*Cross-sectoral climate policies can both promote the increase of forest carbon sinks in the local region and benefit the increase of forest carbon sinks in neighboring regions.*


### 2.3. Moderating Effect of Forest Resource Conservation and Utilization

The conservation and utilization of forest resources are important influencing factors leading to changes in forest carbon sinks. The reduction in deforestation and the increase in forest area promote the significant increase of forest carbon sinks [13,17,18]. In China, local governments have increased forest carbon sinks by implementing large-scale afforestation and reforestation policies [51,52,53,54], and sustainable forest management practices have also contributed to maintaining and increasing forest carbon sinks [55,56,57,58]. Since increasing forest area is a direct way to increase forest carbon sinks [52], provinces with richer forest resources have larger total forest carbon sinks and, therefore, more room to increase forest carbon sinks. Then, provinces with higher forest areas may rely more on their forest resource endowment to achieve the goal of increasing forest carbon sinks [3,59,60], and the role of cross-sectoral climate policies in promoting forest carbon sinks may be weakened. Moreover, in the long term, there is an interaction between forestry industry development and forest carbon sinks [61]. If a region has a well-developed forestry industry, the forestry output of the region usually tends to increase with the increase in forest resources [3], and the forest carbon sinks of the region also tend to increase. The high demand for wood products, however, leads to an increase in wood production, which in turn leads to decreased forest carbon sinks and increased carbon emissions [18]. Thus, the impact of cross-sectoral climate policies on forest carbon sinks in regions with developed forestry industries may be influenced by forestry industry development factors. In summary, forest resource use and conservation behavior may have a moderating effect on the relationship between cross-sectoral climate policies and forest carbon sinks. Accordingly, this paper proposes Hypothesis 3.

**Hypothesis 3:** 
*Forest resource conservation and utilization variables have a moderating effect on the impact of cross-sectoral climate policies on forest carbon sinks.*


## 3. Materials and Methods

### 3.1. Variable Selection and Data Sources

#### 3.1.1. Measurement of Forest Carbon Sinks

Accurate estimation of forest carbon sinks not only helps to explain the imbalance of carbon balance but also promotes the development of forestry carbon sink trading [62]. Currently, there are various methods for measuring forest carbon sinks, which are all applicable but not yet unified [62,63]. Both forest carbon sequestration and forest carbon storage can be used to characterize forest carbon sinks. Among them, forest carbon storage is the reserve of carbon elements in each carbon pool of the forest ecosystem at a certain time, which is the result of the accumulation of the forest ecosystem for many years. Many studies have used forest carbon storage to characterize forest carbon sinks [33,61,64,65], and researched the issue of forest carbon sinks. At present, the methods for accounting for forest carbon storage include two types: one is the sample plot inventory method, which specifically includes biomass, the stockpile method, and the biomass inventory method. The other is the micrometeorological method, which measures the concentration of carbon dioxide by the technique of meteorological principles, which includes the eddy correlation, eddy covariance, relaxed eddy accumulation conversion, and box methods [65,66]. Of these, the forest carbon storage calculated by the accumulation conversion method is very close to the data of the actual monitored carbon storage in China, and the carbon sink data calculated using this method are more detailed and practical [64]. To ensure the forest carbon sink data are comparable among different provinces, forest carbon storage is used as the dependent variable for analysis in this paper. The theoretical model of forest carbon storage is as follows.
(1)$$\{\begin{array}{c} C\left(k+1\right)=C\left(k\right)+G\left(k\right)-W\left(k\right)-L\left(k\right)\\ C\left(k_{0}\right)=C_{0}\\ C\left(k\right)\geq 0,\;\;0\leq L\left(k\right)\leq L{\left(k\right)}_{max} \end{array}$$

In Equation (1), *C*(*k*) is the accumulated carbon stock in forests, *G*(*k*) is the carbon stock for forest growth, *W*(*k*) is the carbon stock of forest dieback, *L*(*k*) is the carbon stock of forest harvesting, and *k* is the year. According to Xi and Li [67], the equation for measuring forest carbon sinks using the stock conversion method is as follows.
(2)$$FCS=V_{F}\delta \rho \gamma +\alpha V_{F}\delta \rho \gamma +\beta V_{F}\delta \rho \gamma$$
where FCS is forest carbon storage, and *V_F_* denotes forest stock volume; *δ* is the coefficient of converting forest stock to biomass stock, namely biomass expansion coefficient; *ρ* is the coefficient of converting forest biomass stock to biotic dry weight, namely bulk density; *γ* is the coefficient of converting biotic dry weight to carbon sequestration, that is, carbon content rate; *α* is the conversion factor of carbon sequestration in understory vegetation, and *β* is the conversion factor of carbon sequestration in forest land. Here, *δ* = 1.90, *ρ* = 0.50, *γ* = 0.50, *α* = 0.195, and *β* = 1.244.

#### 3.1.2. Quantification of Cross-Sectoral Climate Policies

The implementation capacity of climate policy directly impacts the climate policy effects [36]. Given the decentralization of decision-making power and the plurality of interest subjects, however, there may be negative effects among multiple climate policies [21], such as the existence of strong substitutability and the problem of exclusion between policies. Cross-sectoral climate policies are more likely to gain support from multiple sectors in the implementation process to achieve policy effects [20]. For the quantification of climate policies, Peng et al. used the number of policies jointly issued by multiple departments, the number of jointly issued departments, and the level of joint promulgation policies to describe the degree of coordination of government departments [68]. Based on Zheng et al., this paper chose the cumulative number of climate policies jointly issued by multiple departments in the form of accumulation to represent the cross-sectoral coordination level of climate policies [29]. The theoretical model for cross-sectoral climate policy quantification is as follows:(3)$$CSCP\left(k\right)=\overset{k}{\underset{i=1}{\sum }}S_{i},\;S_{i}\geq 0$$

In Equation (3), CSCP(*k*) is the cross-sectoral synergy index of climate policies, that is, the cumulative number of climate policies jointly promulgated by multiple sectors in year *k*, and *S_i_* is the number of climate policies jointly issued by multiple sectors.

#### 3.1.3. Selection of Control and Moderating Variables

The influencing factors of forest carbon sinks involve multi-disciplinary fields and are complex. Based on existing studies, this paper selected both socioeconomic and natural factors as relevant control variables. In terms of socioeconomic aspects, economic growth and urbanization are considered the main drivers of forest carbon sinks [3,19]. Among them, the economic growth using the gross domestic product (GDP) growth rate is said to measure the regional economic development level (gdp). The proportion of the urban population in the total population was selected to characterize the degree of regional urbanization (urban). Meanwhile, socioeconomic factors such as industrial structure (ser), energy consumption structure (ener), land use structure (land), timber harvesting (harv), and forest management (fm) also have an impact on forest carbon sinks [13,17,34,57,69]. In addition, temperature (temp) and precipitation (prec) were selected as control variables for natural factors, since natural factors also have significant effects on forest carbon sinks [13,16,34]. Moreover, considering the competitive benefits of policies, the number of climate policies issued by a single sector (single) was also included as a control variable in the empirical analysis framework. 

For the measurement of forest resource conservation and utilization behavior, many scholars have chosen different indicators according to the needs of specific studies. Liu et al. chose forest fire control (MFI) and disease, pest, and rodent control (MDPR), and timber yield reduction (MEH) to characterize forest management practices [69], and Tong et al. analyzed the effects of forest management practices on forest carbon sinks concerning land use types [70]. Land use change is a key factor in monitoring forest carbon sinks [70,71]. Moreover, the focus of forest management to store carbon is to increase forest area [72], and the change in forest area can reflect the results of forest protection and utilization. Thus, in this paper, forest area (FA) is used as one of the variables to measure the conservation and utilization behavior of forest resources. In addition, the conservation and utilization of forest resources are crucial to achieving the coordinated development of forest resource utilization and the forestry industry in China [73]. It then also becomes important to examine the impact of forest resource utilization from the perspective of forestry industry development. Thus, in this paper, the gross output value of the forest-related industry of the primary forestry industry (PFP) is also used as a variable to characterize the conservation and utilization of forest resources. Table 1 is a summary of the descriptive statistics of the main variables in the paper.

#### 3.1.4. Data Sources and Description

To ensure the completeness and availability of statistical data, the research sample selected in this paper is 30 provinces and cities in China (excluding Tibet and Hong Kong, Macao, and Taiwan), and the research period is from 2007 to 2020. The data to measure forest carbon sinks are from China Forest Inventory Database. Because the forest inventory cycle is five years, the forest carbon sinks are consistent within the same inventory cycle and mainly include municipal government offices, National Development and Reform Commission, Finance Department, Environmental Protection Department, Housing and Urban-Rural Development Department, etc. Considering the completeness and comprehensiveness of policy texts, the website of PKU law (https://www.pkulaw.com/) is the source used for data retrieval. Excluding policy texts with low relevance, a total of 2050 climate policy texts independently or jointly issued by departments were screened out. The data on economic growth, urbanization rate, industrial structure, and land use structure are from the China Statistical Yearbook. The original data on energy consumption structure come from the China Energy Statistical Yearbook. The data on afforestation management, timber harvest, forest area, and the output value of forest-related industries in the primary forestry industry were obtained from the China Forestry and Grassland Statistical Yearbook. The precipitation and temperature data are mainly from the China Meteorological Yearbook, and some missing data are from the statistical yearbooks of various provinces and cities. Since the results of the 10th Forest Inventory and the 2021 China Energy Statistical Yearbook have not yet been published, the data of forest stock and forest area in 2019-2020 and energy consumption in 2020 were predicted using linear regression to balance the panel data. To avoid bias in estimation results due to data instability and variable dimension differences, the FCS, FA, PFP, FM, harv, prec, and temp were log-transformed.

### 3.2. Model Specification

#### 3.2.1. Benchmark Regression Model

Referring to Du et al. (2021) and based on existing data, we built a two-way fixed effect (TWFE) panel model. Based on the above theoretical analysis, the basic model of the impacts of cross-sectoral climate policies on forest carbon sinks was constructed by including the control variables of socioeconomic and natural factors. The model is as follows:(4)$$lnFSC_{it}=\alpha_{0}+\alpha_{1}CSCP_{it}+\alpha_{2}X_{it}+\mu_{i}+\delta_{t}+\varepsilon_{it}$$

In Equation (4), FSC_it_ is the forest carbon storage of province *i* in year *t*. CSCP_it_ is the cross-sectoral synergy index of climate policy of province *i* in year *t*. *X_it_* is the control variable, including economic development (gdp_it_), urbanization (urban_it_), industrial structure (ser_it_), energy consumption structure (ener_it_), land use structure (land_it_), forest management (fm_it_), timber harvesting (harv_it_), precipitation (prec_it_), temperature (temp_it_), and single-sector climate policy (single_it_). *μ_i_* denotes the individual fixed effect of province *i* that does not vary over time. *δ_t_* controls for time-fixed effects. *ε_it_* represents the random disturbance term. *α*_0_ is the intercept term. *α*_1_ is the regression coefficient of cross-sectoral climate policies. *α*_2_ is the regression coefficient of the control variables. 

#### 3.2.2. Spatial Panel Model

It has been shown that forests can exhibit significant regional correlation effects [13,34] and cannot, therefore, meet the basic assumption of traditional econometric studies that samples are independent of each other. To explore the impact of cross-sectoral climate policies on forest carbon sinks, this paper first used exploratory spatial data analysis to test the spatial correlation of forest carbon sinks. We then used spatial econometric models to analyze the impacts of cross-sectoral climate policies on forest carbon sinks.

The spatial correlation of forest carbon sinks can be tested using the global Moran’s I index, and the calculation formula is as follows:(5)$$I=\frac{n\sum_{i=1}^{n}\sum_{j=1}^{n}W_{ij}\left(x_{i}-x\right)\left(x_{j}-x\right)}{\sum_{i=1}^{n}\sum_{j=1}^{n}W_{ij}\sum_{i=1}^{n}{\left(x_{i}-\bar{x}\right)}^{2}}$$
where *I* is the global Moran’s I index, n is the total number of study samples, *x_i_* and *x_j_* are the forest carbon storage of region *i* and region *j*, and $\bar{x}$ is the average of forest carbon storage. *W_ij_* is the spatial weight, and this paper chooses Rook adjacent matrix. That is, if region *i* and region *j* are adjacent, *W_ij_
*= 1, otherwise, *W_ij_
*= 0. Moran’s I index takes values between −1 and 1. A value greater than 0 indicates that the variables have a positive spatial correlation, and vice versa, that the variables have a negative spatial correlation for values less than 0. If Moran’s I index is greater than 0 and the z-values of the normal statistics are all greater than the critical value of 1.96 at the 0.01 level of the normal distribution function, this indicates that the regions have an obvious positive correlation in spatial distribution. 

The degree of agglomeration on forest carbon sinks was tested by the local Moran’s I index, and the calculation formula is as follows:(6)$$I_{i}=\frac{\left(x_{i}-\bar{x}\right)}{S^{2}}\underset{j}{\sum }W_{ij}\left(x_{j}-\bar{x}\right)$$

*I_i_* is the local Moran’s I index, *x_i_* and *x_j_* are the forest carbon storage in regions *i* and *j*, and *W_ij_* is the spatial weight matrix, $\bar{x}$ is the average of forest carbon storage. If *I_i_* > 0, it means that high values are surrounded by high values, or low values are surrounded by low values. If *I_i_* < 0, it means that high values are surrounded by low values, or low values are surrounded by high values.

If there is a significant spatial correlation in forest carbon sinks, it is necessary to use spatial econometric models for exploration. The commonly used spatial econometric models include the spatial error model (SEM), spatial autoregressive model (SAR), and spatial Durbin model (SDM). Among them, the SDM includes both spatial lag and spatial error, which is more general. Thus, the SDM was used to explore the impact of cross-sectoral climate policies on forest carbon sinks, and the constructed model form is as follows.
(7)$$lnFSC_{it}=\alpha_{0}+\rho WFCS_{it}+\varphi_{1}WCSCP_{it}+\alpha_{1}CSCP_{it}+\varphi_{2}WX_{it}+\alpha_{2}X_{it}+\mu_{i}+\delta_{t}+\varepsilon_{it}$$

In Equation (7), ρ is the spatial autoregressive coefficient, W is the spatial weight matrix (the adjacency weight matrix), and $\varphi_{1}$ and $\varphi_{2}$ are the elasticity coefficients of the core explanatory variables and control variables, respectively. To accurately estimate the interrelationship between cross-sectoral climate policies and forest carbon sinks, it is necessary to select the most appropriate model among different types of spatial panel models for estimation.

According to the LM test, only the LM value and the robust LM value of the SAR passed the significance test. Therefore, it is more appropriate to use SAR to analyze the impact of cross-sectoral climate policies on forest carbon sinks. The LM test results are shown in Table 2.

According to the Hausman test results, SAR with both time and space fixed is selected as the optimal model to explore the impact of cross-sectoral climate policies on forest carbon sinks. As a result, the final form of the spatial panel model is constructed considering the spatial spillover effect of forest carbon sinks as follows.
(8)$$lnFSC_{it}=\alpha_{0}+\rho WlnFCS_{it}+\alpha_{1}CSCP_{it}+\alpha_{2}X_{it}+\mu_{i}+\delta_{t}+\varepsilon_{it}$$

In the above equation, lnFSC_it_ represents the forest carbon sinks of province *i* in year *t*. CSCP_it_ is the cross-sectoral synergy index of climate policies in province *i* in year *t*, which is used to characterize the cross-sectoral synergy level of different provinces. *x_it_* is the control variables, including socioeconomic variables and natural factors. *α*_1_ is used to reflect the degree of the direct impact of cross-sectoral climate policies on forest carbon sinks, and it is the direct impact of the explanatory variables and the impact that feeds back into the region by hitting the neighboring regions. *ρ* is the spatial regression coefficient, which indicates the direction and degree of influence of forest carbon sinks in neighboring provinces on forest carbon sinks in this province. *W* is the spatial weight matrix, *α*_2_ is the regression coefficient of control variables, and *ε_it_* is the random disturbance term.

#### 3.2.3. Moderating Effects of Forest Resource Protection and Utilization

The protection and utilization of forest resources are important factors affecting the change of forest carbon sinks. Previous studies have shown that the reduction of deforestation and the increase of forest area will promote the obvious increase of forest carbon sinks [13,17,18]. Moreover, there is an interaction between forestry industry development and forest carbon sinks [61]. Therefore, the protection and utilization of forest resources may have an impact on the relationship between cross-sectoral climate policies and forest carbon sinks. To explore the moderating effect of forest resource conservation and utilization on the cross-sectoral climate policies and forest carbon sinks, the interaction terms of FA, PFP, and CSCP were introduced based on Equation (8). The significance of the coefficient of the multiplicative term was tested to assess the moderating effect of forest resource protection and utilization behavior on the relationship between cross-sector climate policies and forest carbon sinks. The calculation formula is shown in Equation (9).
(9)$$lnFSC_{it}=\alpha_{0}+\rho WlnFCS_{it}+\alpha_{1}CSCP_{it}+\alpha_{2}X_{it}+\alpha_{3}lnM_{it}+\beta CSCP_{it}\times M_{it}+\mu_{i}+\delta_{t}+\varepsilon_{it}$$

$M_{it}\;$is the moderating variable, which includes forest area (FA) and the total output value of forestry-related industry of forestry primary industry (PFP). *A*_3_ is the regression coefficient of the moderating variables, and *β* is the coefficient of the interaction term. The remaining parameters are consistent with those in Equation (8).

## 4. Empirical Results

### 4.1. Spatial Correlation Test

Because it is difficult to separate the forest in terms of geographical distribution, the spatial distribution of forest carbon sinks may also have a strong correlation. In this paper, the global Moran’s I index was used to test the spatial correlation of forest carbon sinks in China, and the local correlation of forest carbon sinks was measured by the local Moran’s I index.

#### 4.1.1. Global Autocorrelation Test

Table 3 shows the calculation results of the global Moran’s I index of forest carbon sinks in various provinces and cities in China. From Moran’s I coefficient and Z value, it can be seen that the forest carbon sinks in China from 2007 to 2020 have a significant positive spatial correlation at the significance level of 1%. In other words, the forest carbon sinks present the spatial distribution characteristics of high–high aggregation and low–low aggregation. Therefore, a spatial panel model was constructed in this paper to analyze the influencing factors and spillover effects of forest carbon sinks.

#### 4.1.2. Local Autocorrelation Test

To further observe the spatial agglomeration characteristics of regional forest carbon sinks, we drew the local Moran’s I index scatter plots of provinces and cities in 2007 and 2020. The first to fourth quadrants in the figure are high–high, low–high, low–low, and high–low agglomeration areas, which represent the local spatial correlation between the forest carbon sinks of a province and other surrounding provinces and cities. Limited by space, the local spatial correlation of forest carbon sinks in various provinces in China is only presented in 2007 and 2020, as shown in Figure 1. It can be seen from the figure that the spatial correlation of forest carbon sinks in most provinces and cities is located in the first and third quadrants. Thus, this further indicates that there is a significant positive spatial spillover effect of forest carbon sinks. Among them, Jilin, Heilongjiang, Inner Mongolia, Guangxi, and other provinces with rich forest resources have the characteristics of high–high agglomeration of forest carbon sinks. The provinces with developed economies but scarce forest resources, such as Shanghai, Beijing, Tianjin, and Hebei, had characteristics of low–low agglomeration of forest carbon sinks. The provinces with the high–high agglomeration of forest carbon sinks are concentrated in southwest and northeast China, which is consistent with the research results of Xue et al. [34].

### 4.2. Impacts of Cross-Sectoral Climate Policies on Forest Carbon Sinks

We more comprehensively explored the impact of cross-sectoral climate policies on forest carbon sinks. Based on the existing research and obtained data, we separately conducted estimations using either the ordinary econometric model or the spatial panel model. First, we use mixed estimates and fixed effects regression without considering spatial spillover effects to test the impact of cross-sectoral climate policies on forest carbon sinks. Second, the space elements are included in the empirical analysis framework. According to the Hausman test results, the spatial lag model with both time and space fixed was used to explore the effect of cross-sectoral climate policies on forest carbon sinks.

#### 4.2.1. Benchmark Regression Results

Stata 17.0 was used to perform benchmark regression according to formula (4), and the regression results are shown in Table 4. Among them, columns (1) and (2) in Table 4 represent the regression results estimated based on mixed estimation and the panel fixed effects, respectively. Table 4 shows that the two methods of estimation of goodness-of-fit coefficient (R^2^) are high, showing that the two methods of estimate that have a good fitting effect. Since the R^2^ of the mixed estimation is 0.814 and the results of the mixed estimation and panel fixed effect regression are consistent, the calculation results of the mixed regression are selected for analysis.

It can be seen from Table 4 that the regression coefficient of the cross-sectoral climate policy index is positive and passes the 1% significance level test. For every percentage point increase in the cross-sectoral climate policy index, forest carbon storage increases by 0.023 percentage points. The results indicate that cross-sectoral climate policies play a significant role in promoting forest carbon sinks, which is consistent with Hypothesis 1.

Regarding control variables, with economic development, urbanization, and traditional energy consumption accounted for, and the annual average temperature, there is a significant negative impact on forest carbon storage. The forest area proportion, wood harvest volume, and forest management behavior had positive and significant effects on forest carbon storage. The above conclusions are consistent with many studies [18,34,40,52]. This result, that the amount of wood harvested has a positive effect on forest carbon storage, is somewhat counterintuitive. Through investigating the reasons, it is possible that carbon-rich tree species can be protected under the forest cutting quota system of deforestation from natural forest to plantation. Moreover, rational planning of forest cutting can effectively improve the growth environment of trees [34], which is conducive to an increase in forest carbon storage.

#### 4.2.2. Regression Results for the Spatial Panel Model

Following the above analysis, we adopt the cross-sector climate policy index as the core explanatory variable and perform regression using the SAR with both time and space fixed. The regression results are shown in column (3) of Table 4. Column (4), column (5), and column (6) in Table 4 decompose the impact of the cross-sectoral climate policies on forest carbon sinks into direct effect, indirect effect, and total effect, respectively.

Firstly, the Log-L and Sigma2 statistics show that the model fits well and has high overall credibility. The spatial lag coefficient ρ is significantly positive at the 1% significance level, indicating that forest carbon sinks have a significant positive spatial correlation. That is, the higher the forest carbon storage in neighboring provinces, the higher the forest carbon storage in this province, which is consistent with the spatial correlation test results of forest carbon sinks as described above. Specifically, for every 1% increase in the forest carbon storage of neighboring provinces and cities, the forest carbon storage of the province will increase by 0.405%. Secondly, when considering both temporal and spatial fixed effects and control variables, climate policies across sectors have significant positive spillover effects on forest carbon sinks in provinces and cities, consistent with Hypothesis 2. It can be seen that the increase of forest carbon sinks is not a single fight of each province or city but needed for strengthening regional cooperation [74,75]. All provinces and cities should work together to achieve the goal of increasing forest carbon sinks and enhancing the ability to cope with climate change.

Direct and indirect effects are used to explain the impact of climate policies on forest carbon sinks across sectors since partial differentials of variable changes can allow the bias of point estimates to be avoided in the analysis of spatial spillovers between regions. In terms of direct effects, the regression coefficients of the climate policy index across sectors are significantly positive. Holding other influencing factors constant, a 1% increase in the intersectoral climate policy index will increase the forest carbon storage of the province by 0.004 percentage points on average. In terms of indirect effect, the regression coefficient of the cross-sectoral climate policy index is still significantly positive, indicating that the cross-sectoral climate policy index has a spatial spillover effect. The improvement of the cross-sectoral climate policy index in neighboring provinces will also contribute to the increase in forest carbon storage in their provinces. There are two main reasons for this analysis: First, in the context of the time when attention is paid to coping with climate change, the improvement of cross-sectoral coordination of climate policy will make climate policy more effective [20,29]. Multi-sectoral climate policies with good results will produce a “demonstration effect” [76], which will promote the surrounding provinces to follow suit and further promote the cross-sectoral coordination of climate policies. In this context, forest carbon sinks can be improved. Second, the forest carbon sinks itself has a positive externality. When provinces and cities increase their forest carbon sinks, they will also promote the increase in forest carbon sinks in neighboring provinces and promote the continuous increase of forest carbon sinks in the region [34,60].

#### 4.2.3. Impact of Forest Resource Conservation and Utilization

The increase in forest carbon sinks in China is an objective fact [3,16,65], and the protection and utilization of forest resources have a significant impact on forest carbon sinks [61,70,77]. Then, the impact of cross-sectoral climate policies on forest carbon sinks may be influenced by relevant factors. Therefore, based on the original model, the interaction terms of forest area, the total output value of forest-related industries of the forestry primary industry, and the cross-sectoral collaboration index of climate policy are introduced. The results of the empirical regression are shown in Table 5. Among them, model (1) and model (2) are the estimation results of the interaction terms of the intersectoral synergy index including the proportion of forest area, the total output value of forest-related industries of the forestry primary industry, and climate policy, respectively. It should be noted that to avoid the problem of multicollinearity, the variable of land use structure (the ratio of forest area to land area, land) in the original control variables was eliminated in model (1), and the remaining control variables are still included in the model.

According to the empirical results, all interaction terms are significantly negative at the significance level of 1%. First, forest areas will weaken the positive impact of cross-sectoral climate policies on forest carbon storage. The coefficient of forest area, since it is positive, indicates that there is a clear substitution relationship between forest area and climate policies across sectors. Therefore, for provinces with small forest areas, such as Shanghai and Beijing, the positive impact of cross-sectoral climate policies on forest carbon storage is more obvious. With the increase in forest area, forest resources become more abundant, and the positive effect of cross-sectoral climate policies on forest carbon storage gradually decreases. The reason is that the abundance of forest carbon sinks largely depends on the forest area of a region [78]. In other words, provinces with rich forest resources tend to have higher forest carbon storage [3,59,60,79]. For these regions, the increase in forest area plays a major role in the increase of forest carbon storage [54,54], which will weaken the positive impact of cross-sectoral climate policies on forest carbon storage. Second, the total output value of the forest-related industry of the forestry primary industry will also inhibit the positive impact of cross-sectoral climate policies on forest carbon storage, and there is an obvious substitution effect between the total output value of the forest-related industry of the forestry primary industry and cross-sectoral climate policies. This indicates that the provinces with the lower development level of the forestry primary industry experience more significant positive effects on forest carbon storage resulting from cross-sectoral climate policies. Provinces with a more developed forestry industry tend to have rich forest resources [80], and the forestry output value in this region usually shows a trend of increasing with the increase in forest resources [3], and the forest carbon sinks in this region also show an increasing trend. As a result, forest carbon sequestration in this region is more significantly affected by its forestry industries, such as in the case of Guangxi and Zhejiang provinces, which are located in the collective forest areas of South China.

### 4.3. Robustness Test

#### 4.3.1. Replacement of Dependent Variable

To more fully reflect the relationship between climate policy coordination and forest carbon storage, the increase in and growth rate of forest carbon storage were used instead of forest carbon storage for the robustness test. Among them, column (1) takes the increment of forest carbon storage as the dependent variable, referring to Yang et al. [52]. It is also assumed that the annual increase in forest carbon storage during the forest resource inventory cycle is one-fifth of the total increase in forest carbon storage. Column (2) takes the growth rate of forest carbon storage as the dependent variable. The growth rate of forest carbon storage was based on the sixth National Forest Resources Inventory (1999–2003), and the growth rate of forest carbon storage in each province and city from 2007 to 2020 was then calculated. The increase in forest carbon storage and the growth rate of forest carbon storage cannot completely pass the test of spatial correlation. Therefore, the panel fixed effect model is used for regression, and the regression results are shown in Table 6.

The model results show that the increase in forest carbon storage (lnFCSI) and the growth rate of forest carbon storage (lnFCSG) both pass the significance test, with a positive regression coefficient. This indicates that cross-sectoral climate policies have significant positive effects on the increase in forest carbon storage and the growth rate of forest carbon storage, which is consistent with the previous results. This suggests that cross-sectoral climate policies have a robust positive effect on forest carbon sinks.

#### 4.3.2. Replacing the Weight Matrix

For the spatial econometric model, different spatial weight matrices will eventually produce different results [81]. Therefore, different spatial weight matrices are considered for robustness testing [16]. To test the robustness of the results, the geographical distance between provinces was used to construct the geographical weight matrix. LeSage and Polasek incorporated the transportation network into the construction of the spatial weight matrix [82]. Based on the practice of Shao et al., the nearest highway mileage between the regional capital and the capital of j province is selected as the geographical distance between provinces [83], where *d_ij_* is the geographical distance between province i and province j. The basic form is as follows:$$W_{ij}^{d}=\{\begin{array}{c} 1/d_{ij},i\neq j\\ 0,\;i=j \end{array}$$

Consider that the development of forest carbon sinks is influenced not only by geographical characteristics but also by socioeconomic characteristics. Therefore, the economic weight matrix is constructed from the perspective of economic attributes. Referring to the practice of Shao et al. and Zhou and Li [83,84], the annual arithmetic mean of GDP per capita of different provinces in the sample period was used to construct the spatial weight matrix of the cross-section. Among them, ${\bar{X}}_{i}$ and ${\bar{X}}_{j}$ refer to the arithmetic average of per capita GDP in the sample periods of provinces i and j, respectively. The basic form is as follows:$$W_{ij}^{e}=\{\begin{array}{c} 1/\left|\bar{X_{i}}-{\bar{X}}_{j}\right|,i\neq j\\ 0,\;i=j \end{array}$$

The nested weight matrix not only considers the spatial impact of geographical distance but also reflects the fact that there are regional spillovers and radiation effects of economic factors [83,85]. It can more accurately analyze the spatial impact of climate policy coordination on forest carbon sinks. Combining the spatial weight matrix of geographic distance and economic distance, the nested spatial matrix of geographic and economic distance is constructed. The basic form is as follows:$$W_{ij}^{de}=\{\begin{array}{c} \left(1-\varphi \right)W_{ij}^{d}+\varphi W_{ij}^{e},\;\;i\neq j\\ 0,\;\;i=j \end{array}$$

Here, the value of *φ* ranges from 0 to 1. The closer *φ* is to 0, the more important the spatial weight matrix is to the geographical distance between different provinces. When *φ* is closer to 1, the spatial weight matrix focuses more on the economic distance between different provinces. To simplify the analysis, *φ* is set to 0.5 in this paper.

To test the robustness of the empirical results above, the above geographical distance weight matrix, economic distance weight matrix, and nested weight matrix are each used for regression, and the regression results are shown in Table 6 for Model (3), Model (4), and Model (5), respectively. It is not difficult to find that the regression coefficients of the cross-sectoral climate policy (CSCP) are positive under the three weight matrices. All are significant at the significance level of 1%, which indicates the high robustness of the empirical results of the model.

## 5. Discussion

To expedite the realization of carbon peak and carbon neutrality, local governments should not only limit carbon emissions but also pay attention to increasing carbon sinks [75]. Nature-based approaches are currently the most cost-effective way to increase carbon sinks [86]. Fang et al. showed that the annual carbon sequestration rate of the forest was significantly higher than that of grassland, shrubs, and crops [51]. Thus, it is considered that forests play an important role in the global carbon cycle [16] and are valued in climate change mitigation due to their high carbon storage and productivity [11]. Through empirical tests, it is found that forest carbon sinks have a significant spatial correlation both at the national level and at the provincial level [13,34]. Liu et al. also took counties in Shaanxi Province of China as research samples and found that forest carbon storage has a significant positive spatial correlation in geographic space [75]. By examining the spatial correlation of forest carbon sinks in different provinces and cities, we also find a significant positive spatial correlation of forest carbon sinks in China from 2007 to 2020. This is consistent with the conclusions of existing studies on spatial correlation tests for forest carbon sinks.

In contrast to the existing literature, this paper not only examined the relationship between cross-sectoral climate policies and forest carbon sinks from the perspective of traditional measurement methods but also the spatial spillover effects using spatial measurement methods. The empirical results show that cross-sectoral climate policies have a significant positive impact on forest carbon storage, with significant positive spatial spillover effects. The improvement of cross-sectoral coordination in climate policy will make climate policy more effective [20,29]. In the context of regional integration, there are “demonstration effects” among regions, and cross-sectoral climate policies with good results will be regarded as “positive cases”. Surrounding regions will follow suit and develop cross-sectoral climate policies and management measures that are commensurate with their development realities to better achieve their carbon peak and carbon neutrality targets. The same is true at the national level. Sweden’s climate policy fully considers cross-sector equity and achieves a win–win situation between climate and economy, which makes Sweden a leading model in the field of climate policy in the EU and among other OECD countries or regions [76]. Moreover, the forest carbon sinks itself has a positive externality [13,34,75]. While improving its forest carbon sinks, provinces and cities will promote the improvement of forest carbon sinks in neighboring provinces [60], thereby promoting the continuous improvement of forest carbon sinks in the region. In addition, this paper discusses the mechanism of the impact of forest resource conservation and utilization on the relationship between cross-sectoral climate policies and forest carbon sequestration. The moderating role of forest resource protection and use in cross-sectoral climate policy was verified. The results show that both forest area and the forest-related industry output value of forestry primary industry will weaken the positive impact of cross-sectoral climate policies on forest carbon storage. Previous studies have shown that provinces with rich forest resources tend to have higher forest carbon storage [3,59,60,79], and the forest output value usually shows an increasing trend with the increase of forest resources [3,80]. As a result, the positive impact of cross-sectoral climate policies on forest carbon storage is not obvious for provinces with abundant forest resources and developed forestry industries.

Compared with the existing research, this paper has some innovations in the research content. There is no denying, however, that there are still some shortcomings in this work. First, this study only takes the number of climate policies jointly issued by various provinces and departments as a proxy variable when quantifying cross-sectoral climate policies. Thus, policy effectiveness and other issues are not comprehensively considered. Second, the influencing factors of forest carbon sinks are quite complex. In this paper, only some important indicators are selected as control variables in existing studies, and the control variable system still needs to be improved. Finally, the impact of cross-sectoral climate policies on forest carbon sinks is discussed only from the perspective of local governments. The impact of the interaction between central and local governments is not considered. In later studies, it is necessary to incorporate cross-sectoral climate policies at the national level into the analytical framework.

## 6. Conclusions

Based on China’s provincial panel data from 2007 to 2020, this paper analyzes the impact of cross-sectoral climate policies on forest carbon sinks by using common panel regression models and spatial lag models. The moderating effect of forest resource conservation and utilization on the relationship between cross-sectoral climate policies and forest carbon sinks was also explored. Our main findings are summarized below. First of all, the forest carbon storage of provinces and cities in China has significant spatial autocorrelation and, on the whole, shows high–high aggregation in areas with abundant forest resources and low–low aggregation in areas with relatively poor forest resources. Secondly, the estimation results of baseline regression and spatial lag model show that the cross-sectoral climate policies not only significantly promote forest carbon storage in the region but also have a positive impact on the forest carbon storage in the neighboring region. Further studies showed that forest resource protection and utilization variables would weaken the positive impact of cross-sectoral climate policies on forest carbon sinks. The positive impact of cross-sectoral climate policies on forest carbon sinks is more significant in provinces with smaller forest areas and lower levels of forestry industry development.

Cross-sectoral coordination of climate policies is the “main theme” of future climate policies and plays an important role in national and regional climate governance. Based on the empirical model results, the policy implications of this paper are reflected in the following aspects:

First, when different provinces increase forest carbon sinks, they should clarify the actual and potential levels of forest carbon sinks and the influencing factors of forest carbon sinks. Different provinces should be encouraged to strengthen linkages and cooperation in increasing forest carbon sinks, clarifying the responsibilities and division of labor for each province in the region, and giving full play to the positive spillover effect of forest carbon sinks. For example, for the provinces with limited development space for forest carbon sinks, the neighboring provinces with greater potential for increasing forest carbon sinks, should be supported with funds and talents.

Second, there should be a focus on the positive role of cross-sectoral climate policies in increasing forest carbon sinks. All provinces and municipalities need to pay attention to the significant positive spillover effects of cross-sectoral climate policies on forest carbon sinks. Central government departments should actively guide provincial government departments to strengthen coordination among various departments in formulating and implementing climate policies. Because forest area and forestry output value can inhibit the positive effect of cross-sectoral climate policies on forest carbon sinks, provinces with rich forest resources and developed forestry industries should pay more attention to the impact of cross-sectoral climate policies on forest carbon sinks. Cross-sectoral climate policies should be promoted through strengthening cross-sectoral cooperation to increase forest carbon sinks.

Third, tackling climate change has never been the responsibility of a single government department but, rather, the common responsibility and obligation of different government departments. Intersectoral climate policy coordination has a significant sink enhancement effect; that is, climate policy coordination can effectively increase forest carbon storage in a region and its neighboring regions. Thus, to deal with climate change, especially in the aspect of increasing forest carbon sinks, government departments at all levels should pay more attention to the importance of interdepartmental climate policy coordination when implementing climate policies. The forestry sector should not be regarded as the only department responsible for increasing forest carbon sinks but should be combined with the role positioning of different government departments in increasing forest carbon sinks toward determining the responsibilities and obligations of different government departments in increasing forest carbon sinks, achieving the main goal of increasing forest carbon sinks, and improving the ability to respond to climate change.

## Figures and Tables

**Figure 1 ijerph-19-14334-f001:**
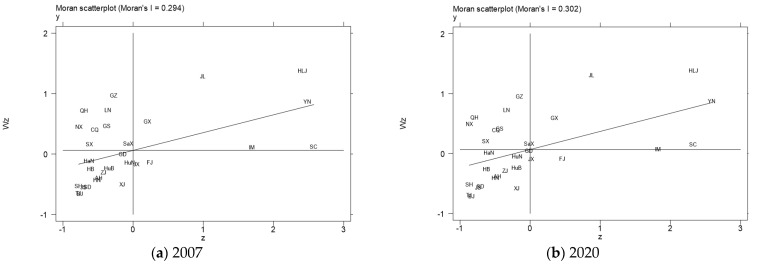
The local Moran scatter plots of forest carbon sinks in China in 2007 (**a**) and 2020 (**b**).

**Table 1 ijerph-19-14334-t001:** Descriptive statistics of main variables.

Variables	Observations	Mean	Std. Dev.	Min	Max
FCS	420	52,515.17	62,363.76	116.95	245,145.35
CSCP	420	6.05	6.45	0.00	35.00
FA	420	691.28	605.92	1.89	2614.85
PFP	420	515.48	462.91	2.14	2403.27
gdp	420	109.46	3.68	95.00	119.20
urban	420	56.39	13.44	28.24	89.60
ser	420	44.68	8.79	15.80	61.50
ener	420	41.27	15.34	1.22	72.42
land	420	31.57	17.67	2.91	65.45
fm	420	174,736.11	174,415.43	1117.00	907,398.00
harv	420	76.46	448.73	0.00	3600.00
prec	420	990.20	728.12	108.60	11,127.00
temp	420	14.04	5.42	2.30	25.70
single	420	4.36	3.01	0.00	17.00

Note: gdp, urban, ser, ener, and land indicators are all ratio indicators, and their units are all in %.

**Table 2 ijerph-19-14334-t002:** Lagrange multipliers and their robust form test results.

Variables	LMERR	R-LMERR	LMLAG	R-LMLAG
Statistical values	0.864	0.006	30.552	29.694
P	0.353	0.940	0.000	0.000

**Table 3 ijerph-19-14334-t003:** Global Moran’s I index of forest carbon sinks.

	2007	2012	2017	2020
coefficient	0.294 ***	0.296 ***	0.301 ***	0.302 ***
Z	2.779	2.782	2.819	2.826

Note: *** indicate significance at 1% levels.

**Table 4 ijerph-19-14334-t004:** Estimation results of benchmark regression and spatial panel model.

Variable	(1)	(2)	(3)	(4)	(5)	(6)
	POLS	FE	SAR	LR_Direct	LR_Indirect	LR_Total
CSCP	0.023 *** (0.008)	0.015 *** (0.002)	0.004 ** (0.002)	0.004 ** (0.002)	0.002 ** (0.001)	0.006 ** (0.003)
gdp	−0.054 *** (0.013)	−0.005 * (0.003)	0.010 *** (0.003)	0.010 *** (0.003)	0.007 ** (0.003)	0.017 *** (0.005)
urban	−0.045 *** (0.004)	−0.001 (0.002)	−0.022 *** (0.003)	−0.022 *** (0.003)	−0.014 *** (0.004)	−0.037 *** (0.006)
ser	0.002 (0.006)	−0.009 *** (0.002)	−0.003 ** (0.002)	−0.004 ** (0.002)	−0.002 * (0.001)	−0.006 ** (0.003)
ener	−0.007 * (0.004)	0.001 (0.002)	0.001 (0.001)	0.001 (0.001)	0.001 (0.001)	0.002 (0.002)
land	0.050 *** (0.003)	0.029 *** (0.003)	0.016 *** (0.002)	0.017 *** (0.003)	0.011 *** (0.003)	0.027 *** (0.005)
af	0.317 *** (0.041)	−0.029 ** (0.010)	−0.007 (0.009)	−0.008 (0.010)	−0.005 (0.006)	−0.013 (0.016)
harv	0.145 *** (0.014)	−0.012 ** (0.005)	−0.011 *** (0.004)	−0.011 *** (0.003)	−0.007 ** (0.003)	−0.019 *** (0.006)
prec	−0.068 (0.104)	−0.007 (0.033)	−0.019 (0.024)	−0.018 (0.025)	−0.011 (0.017)	−0.029 (0.041)
temp	−0.942 *** (0.094)	−0.106 (0.073)	−0.061 (0.055)	−0.061 (0.057)	−0.039 (0.038)	−0.100 (0.094)
single	0.011 (0.012)	0.003 (0.002)	0.002 (0.002)	0.002 (0.002)	0.001 (0.001)	0.003 (0.003)
Constant	15.570 *** (1.720)	10.710 *** (0.561)				
Spatial fixed effects	NO	YES	YES			
Time fixed effects	NO	YES	YES			
ρ			0.405 *** (0.060)			
sigma2			0.007 *** (0.001)			
Log-L			429.890			
R-squared	0.814	0.711	0.467			
Observations	420	420	420			

Note: Standard errors are in parentheses. *, **, *** indicate significance at 10%, 5%, and 1% levels, respectively.

**Table 5 ijerph-19-14334-t005:** The results of the mediating effect model.

Variable	(1)	(2)
CSCP	0.046 *** (0.005)	0.105 *** (0.011)
FA	0.128 ** (0.051)	
EC		0.036 *** (0.013)
CSCP*FA	−0.008 *** (0.001)	
CSCP*EC		−0.006 *** (0.001)
Control variables	YES	YES
Spatial fixed effects	YES	YES
Time fixed effects	YES	YES
ρ	0.334 *** (0.057)	0.421 *** (0.057)
sigma2	0.005 *** (0.000)	0.006 *** (0.000)
R-squared	0.384	0.380
Log-L	509.002	472.791
Observations	420	420

Note: Standard errors are in parentheses. **, *** indicate significance at 5%, and 1% levels, respectively.

**Table 6 ijerph-19-14334-t006:** The results of the robustness test.

	Replace the Dependent Variable		Replace the Weight Matrix		
	(1)	(2)	(3)	(4)	(5)
Variables	LnFCSI	LnFCSG	W^d^	W^e^	W^de^
CSCP	0.019 *** (0.006)	10.810 *** (1.264)	0.006 *** (0.002)	0.008 *** (0.002)	0.007 *** (0.002)
Control variables	YES	YES	YES	YES	YES
Spatial fixed effects	YES	YES	YES	YES	YES
Time fixed effects	YES	YES	YES	YES	YES
ρ			0.252 *** (0.142)	−0.231 *** (0.062)	−0.306 ** (0.118)
sigma2			0.008 *** (0.001)	0.008 *** (0.001)	0.008 *** (0.001)
Log-L			412.502	417.406	414.019
R-squared	0.670	0.459	0.496	0.546	0.543
Observations	420	420	420	420	420

Note: Standard errors are in parentheses. **, *** indicate significance at 5%, and 1% levels, respectively.

## Data Availability

Not applicable.

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
