# Peer review of "Impacts of Cross-Sectoral Climate Policy on Forest Carbon Sinks and Their Spatial Spillover: Evidence from Chinese Provincial Panel Data"

_ijerph, 2022, doi:10.3390/ijerph192114334_

Round 1

Reviewer 1 Report

The subject matter (presented in the paper) is important and current. It concerns correlation / interdependence of cross-sectoral climate polices (in regions) with forests' ability to storage carbon. Basing on the data of 30 provinces and cities in China from 2007-2020, the Authors made the successful attemt elaborating an interesting and partly novel methodic approach, then testing it to explain some phenomena in this field of research.

Presented methodic approach is the most valuable impact of the paper. Methodic construction consits, among others, of a benchmark regression model, a spatial panel model and special selection of variables - to analyze the correlation between the impact of cross-sectoral climate policies and  forest carbon sinks.

Hypothesis (formulated by Authors) are rather obvious  - I rate them average (results of research are easy to predict before research).

Detailed comments:

Ad ABSTRACT. Lines 19-21. What contents did the Authors want to express? The meaning is not clear. Is there a kind of contradiction? "Utilization of forest resources" makes different impact (oposite one), than "protection of forests". Construction of the sentence - shoul it be corrected?

Ad MATERIALS and METHODS. Lines 464-465: something is wrong there (the sentence is constructed badly).

In general, the paper is very interesting and written well. It is worth publishing - mainly because of interesting and partly novel methodic approach. 

Author Response

Dear Reviewer,

On behalf of all authors, we appreciate your positive and constructive comments and suggestions on our manuscript entitled “Impacts of cross-sectoral climate policy on forest carbon sinks and their spatial spillover: Evidence from Chinese provincial panel data” (ijerph-1956215). Those comments are all valuable and very helpful for revising and improving our paper, as well as the important guiding significance to our researches.

We have studied the comments carefully and have made correction which we hope meet with approval. Revised portions are marked in red in the paper. The main corrections in the paper and the responds to your comments are as flowing.

Comment 1. Lines 19-21. What contents did the Authors want to express? The meaning is not clear. Is there a kind of contradiction? "Utilization of forest resources" makes different impact (opposite one), than "protection of forests". Construction of the sentence - should it be corrected?

Reply1: Thank you for your comments. We apologize for any confusion caused by the original expression. The revised contents are as follows:

Lines 20-22: Further analysis shows that for provinces with less developed forestry industry and small forest areas, the positive impact of cross-sectoral climate policies on forest carbon sinks is more obvious.

Comment 2. Lines 464-465: something is wrong there (the sentence is constructed badly).

Reply2: We feel really sorry for this mistake, and we have revised the expression. The revised contents are as follows:

Lines 473-476: We more comprehensively explored the impact of cross-sectoral climate policies on forest carbon sinks. Based on the existing research and obtained data, we separately conducted estimations using either the ordinary econometric model or the spatial panel model.

We tried our best to improve the manuscript and made some changes in the manuscript. These changes will not influence the content and framework of the paper. And here we did not list the changes but marked in red in revised paper. We appreciate for Reviewers’ warm work earnestly, and hope that the correction will meet with approval. Once again, thank you very much for your comments and suggestions.

Thank you and best regards.

Sincerely,

The Authors

Reviewer 2 Report

Thanks for invitation to review the following manuscript.

The manusciprt is much suitable to the publication and could be accepted.

From my side following are the suggestion.

## Don’t discuss the routine things it should be based on the facts Cross-sectoral climate policies can reduce the extra costs of climate policy and expedite the 13 achievement of climate targets for carbon reduction. Thus, based on the panel data of 30 provinces 14 and cities in China from 2007 to 2020,

##[3,5,13,18-19] use most relevant ref.?

##Check the equation 1 and second ?(?) ≥ 0, 0 ≤ ?(?) ≤ ?(?)??

## remove the digit after point 2  like this 245145.35

## not visible Figure 1. The local Moran scatter plots of forest carbon sinks in China (a)2007 and (b)2020

 Rest things are ok

Author Response

Dear Reviewer,

On behalf of all authors, we appreciate your positive and constructive comments and suggestions on our manuscript entitled “Impacts of cross-sectoral climate policy on forest carbon sinks and their spatial spillover: Evidence from Chinese provincial panel data” (ijerph-1956215). Those comments are all valuable and very helpful for revising and improving our paper, as well as the important guiding significance to our researches.

We have studied the comments carefully and have made correction which we hope meet with approval. Revised portions are marked in red in the paper. The main corrections in the paper and the responds to your comments are as flowing.

Comment 1. Don’t discuss the routine things it should be based on the facts. Cross-sectoral climate policies can reduce the extra costs of climate policy and expedite the 13 achievement of climate targets for carbon reduction. Thus, based on the panel data of 30 provinces 14 and cities in China from 2007 to 2020.

Reply1: Thank you for your comments. According to your suggestions, we have revised this part of the abstract. The revised contents are as follows:

Lines 13-14: Forest carbon sinks play an important role in addressing climate change, but there are few studies focusing on forest carbon sinks and cross-sectoral climate policies.

Comment 2. [3,5,13,18-19] use most relevant ref.?

Reply2: What you said is very reasonable. This sentence in the original text is a relatively general discussion, so there are many references. Based on your comments, we have deleted the references that are not strongly related to the argument.

Comment 3. Check the equation 1 and second ?(?) ≥ 0, 0 ≤ ?(?) ≤ ?(?)??x

Reply3: Thank you for your comments. According to the problem you mentioned, we have checked the equation carefully.

Comment 4. remove the digit after point 2, like this 245145.35

Reply4: According to your suggestion, we have changed the numbers in Table 1 to keep two decimal places. In order to show the significance of the empirical results more clearly, the numbers in the empirical section of this article are retained three decimal places.

Comment 5. not visible Figure 1. The local Moran scatter plots of forest carbon sinks in China (a)2007 and (b)2020

Reply5: We agree with you very much. We have modified the picture and replaced it with the high-definition picture.

We tried our best to improve the manuscript and made some changes in the manuscript. These changes will not influence the content and framework of the paper. And here we did not list the changes but marked in red in revised paper. We appreciate for Reviewers’ warm work earnestly, and hope that the correction will meet with approval. Once again, thank you very much for your comments and suggestions.

Thank you and best regards.

Reviewer 3 Report

I have inserted my suggestions and queries in the pdf file with the editing tool which is attached 

Author Response

Dear Reviewer,

On behalf of all authors, we appreciate your positive and constructive comments and suggestions on our manuscript entitled “Impacts of cross-sectoral climate policy on forest carbon sinks and their spatial spillover: Evidence from Chinese provincial panel data” (ijerph-1956215). Those comments are all valuable and very helpful for revising and improving our paper, as well as the important guiding significance to our researches.

We have studied the comments carefully and have made corrections which we hope meet with approval. Revised portion are marked in red in the paper. The main corrections in the paper and the responds to your comments are as flowing.

Comment 1. check the format style of sub heading

Reply1: Thank you for your comments. We have checked the sub-title format of the paper according to the submission style of the paper.

Comment 2. Table 1 is not referred in the text. In some variables the Std. Dev values are greater than the mean values. Why?

Reply2: Thank you very much for your comments. First, the introduction of Table 1 in this paper only summarizes the descriptive analysis of the main variables in the model. We refer to the studies of Zhao et al.(2022) and Xia et al.(2022) that have been published in this journal and do not carry out too much descriptive analysis of the data. We sincerely hope that you can accept this practice. Second, among the main variables, the standard deviation of forest carbon sink and other variables is greater than the average because the data cover a long time and a wide area, so the degree of dispersion of the data is relatively large. Therefore, when building the model, the logarithm of the relevant variables was taken, and the mean value of the data was greater than the standard deviation.

Comment 3. Figure quality is not up to the mark. Clear figure should be used. Figure title location is also not suitable.

Reply3: Thank you for your comments. First, we modified the original image and replaced it with a higher definition image. Second, We also changed the title of the graph.

We appreciate for your warm work earnestly, and hope that the correction will meet with approval. Once again, thank you very much for your comments and suggestions.

Thank you and best regards.

Sincerely,

The Authors

Reviewer 4 Report

This paper is a well-researched study on the impact of cross-sectoral climate policy on forest carbon sinks. In this paper, some credible scientific evidences, based on statistical analysis methods, are presented to support the hypothesis. The paper is scientifically sound and should obtain broad international interest.

The manuscript is clearly laid out and all the key elements are present. The abstract and key words are appropriate. The applied methodology is quite new and original in air pollution science. Its presentation and explanation is accurate. Finally, the results are well explained and the claims in the discussion and conclusions are well supported by them.

All the illustrations are clearly and suitably captioned.

The manuscript can be accepted for publication after a minor revision. My suggestions for revision are as follows:

1. The basic model of the impact of cross-sectoral climate policies on forest carbon sinks was constructed in Eq. (4), Eq. (7)-(9). What is the basis for selecting such a form of benchmark regression model? Need more explanation.

2. The implementation effect of policies is often delayed. And they are quite "time lag”. The authors neglect this factor. The time lag effect of policies should be reflected in the model. Thus, the author's model may need improvement. The author should strengthen the discussion about the time lag effect of policies.

Author Response

Dear Reviewer,

On behalf of all authors, we appreciate your positive and constructive comments and suggestions on our manuscript entitled “Impacts of cross-sectoral climate policy on forest carbon sinks and their spatial spillover: Evidence from Chinese provincial panel data” (ijerph-1956215). Those comments are all valuable and very helpful for revising and improving our paper, as well as the important guiding significance to our researches.

We have studied the comments carefully and have made corrections which we hope meet with approval. And we have corrected the English language and style in this article. It's worth noting that the revised portions are marked in red in the paper. The main corrections in the paper and the responses to your comments are as flowing.

Comment 1. The basic model of the impact of cross-sectoral climate policies on forest carbon sinks was constructed in Eq. (4), Eq. (7)-(9). What is the basis for selecting such a form of benchmark regression model? Need more explanation.

Reply1: Thank you for your comments. According to your suggestion, we have added relevant expressions and references in the part of building the model. The additions are as follows:

Lines 348-351: Referring to Du et al. (2021) and based on existing data, we built a two-way fixed effect (TWFE) panel model. And based on the above theoretical analysis, a basic model of the impacts of cross-sectoral climate policies on forest carbon sinks was constructed by including the control variables of socioeconomic and natural factors.

Lines 421-427: The protection and utilization of forest resources are important factors affecting the change of forest carbon sinks. Previous studies have shown that the reduction of deforestation and the increase of forest area will promote the obvious increase of forest carbon sink [13,17-18]. Moreover, there is an interaction between forestry industry development and forest carbon sinks [61]. Therefore, the protection and utilization of forest resources may have an impact on the relationship between cross-sectoral climate policies and forest carbon sinks.

Comment 2. The implementation effect of policies is often delayed. And they are quite "time lag”. The authors neglect this factor. The time lag effect of policies should be reflected in the model. Thus, the author's model may need improvement. The author should strengthen the discussion about the time lag effect of policies.

Reply2: We quite agree with your opinion, and we have taken this question into consideration when building the model. When quantifying cross-sectoral climate policies, we used the practice of Zheng et al. (2021) to select the cumulative number of climate policies jointly issued by multiple sectors in the form of accumulation to characterize the cross-sectoral coordination level of climate policies. This quantitative method takes into account not only the immediate impact of the policy release, but also the time-lag effect of the policy. Therefore, the lag effect of the policy is not considered in the following paper. I hope this is acceptable to you.

We appreciate for your warm work earnestly, and hope that the correction will meet with approval. Once again, thank you very much for your comments and suggestions.

Thank you and best regards.

Sincerely,

The Authors
